# Sphingomyelin Depletion from Plasma Membranes of Human Airway Epithelial Cells Completely Abrogates the Deleterious Actions of *S. aureus* Alpha-Toxin

**DOI:** 10.3390/toxins11020126

**Published:** 2019-02-20

**Authors:** Sabine Ziesemer, Nils Möller, Andreas Nitsch, Christian Müller, Achim G. Beule, Jan-Peter Hildebrandt

**Affiliations:** 1University of Greifswald, Animal Physiology and Biochemistry, Felix Hausdorff-Straße 1, D-17489 Greifswald, Germany; sabine.ziesemer@uni-greifswald.de (S.Z.); nils.moeller@uni-greifswald.de (N.M.); an124100@uni-greifswald.de (A.N.); christian.mueller@uni-greifswald.de (C.M.); 2Department of Otorhinolaryngology, University Hospital, Münster, Germany and Department of Otorhinolaryngology, Head and Neck Surgery, Greifswald University Hospital, D-17489 Greifswald, Germany; achimgeorg.beule@ukmuenster.de

**Keywords:** sphingomyelin, airway epithelial cells, cell physiology, *Staphylococcus aureus*, alpha-toxin

## Abstract

Interaction of *Staphylococcus aureus* alpha-toxin (hemolysin A, Hla) with eukaryotic cell membranes is mediated by proteinaceous receptors and certain lipid domains in host cell plasma membranes. Hla is secreted as a 33 kDa monomer that forms heptameric transmembrane pores whose action compromises maintenance of cell shape and epithelial tightness. It is not exactly known whether certain membrane lipid domains of host cells facilitate adhesion of Ha monomers, oligomerization, or pore formation. We used sphingomyelinase (hemolysin B, Hlb) expressed by some strains of staphylococci to pre-treat airway epithelial model cells in order to specifically decrease the sphingomyelin (SM) abundance in their plasma membranes. Such a pre-incubation exclusively removed SM from the plasma membrane lipid fraction. It abrogated the formation of heptamers and prevented the formation of functional transmembrane pores. Hla exposure of rHlb pre-treated cells did not result in increases in [Ca^2+^]_i_, did not induce any microscopically visible changes in cell shape or formation of paracellular gaps, and did not induce hypo-phosphorylation of the actin depolymerizing factor cofilin as usual. Removal of sphingomyelin from the plasma membranes of human airway epithelial cells completely abrogates the deleterious actions of *Staphylococcus aureus* alpha-toxin.

## 1. Introduction

Airway epithelia form major barriers between inhaled air and the internal space of the body [1]. In vivo, respiratory epithelia are covered by a thick mucus layer. Inhaled microorganisms and other particles stick to that mucus layer and are removed from the airways by the ciliary activity in the periciliary liquid (mucociliary clearance) [2]. Therefore, it is unlikely that inhaled bacteria like the human commensal and opportunistic pathogen *Staphylococcus aureus* (*S. aureus*) readily come into direct contact with the apical surfaces of epithelial cells. However, when the mucociliary clearance is attenuated (as in bedridden or immune deprived patients, or patients with virus infections or cystic fibrosis) bacteria may reach critical densities in the mucus layer and start to secrete soluble virulence factors.

Virulence factors play a central role in the pathogenicity of *S. aureus* [3]. Secreted soluble virulence factors like alpha-toxin (hemolysin A, Hla) may diffuse through the mucus layer and reach the apical surfaces of the epithelial cells [4]. The assumption that Hla may play a role in the onset of *S. aureus* lung infection is supported by the findings of pneumonia patients having generated antibodies against Hla [5,6] and by animals being protected from developing *S. aureus*-mediated pneumonia when vaccinated against Hla [7].

Alpha-toxin is a pore-forming bacterial toxin [8]. It is lytic to red blood cells [9] and toxic to a wide range of mammalian cells [10]. Hla pores increase the membrane permeability for ATP [11,12] and cations like calcium, potassium, and sodium [13,14,15,16,17]. Cation-entry depolarizes the membrane potential in airway epithelial cells and enhances phosphorylation of p38 MAP kinase [17]. Additionally, Hla induces alterations in cell shape by remodeling the actin cytoskeleton and disrupts cell-matrix adhesions in human airway epithelial cells [18,19]. Actin remodeling seems to be mediated by hypo-phosphorylation and activation of cofilin [19], an actin depolymerizing factor [20].

Hla is secreted by the bacteria as a water-soluble 33 kDa monomer that binds to plasma membranes (PM) of eukaryotic host cells. Seven Hla monomers form a non-lytic heptameric pre-pore [21]. Subsequently, all seven subunits simultaneously unfold their pre-stem domains, which are then inserted into the lipid bilayer and form a cylindrical transmembrane pore [4,22,23].

Binding of Hla monomers to the plasma membranes of host cells may be facilitated by protein-receptors. Potential Hla receptors are the metalloproteinase ADAM10 (a disintegrin and metalloproteinase 10) [24,25], which is also expressed in the plasma membranes of airway epithelial model cells [26], alpha5beta1-integrin [27,28], rabbit erythrocyte band 3, or caveolin [29,30]. Specific toxin binding through Hla receptors appears to be important at low concentrations of Hla [8,31]. At higher concentrations, Hla is able to bind directly to PM lipids [31], probably by interacting preferentially with phosphocholine headgroups of sphingomyelin (SM) and phosphatidylcholine (PC) [32,33,34,35].

SM and PC form clusters with cholesterol in so-called lipid rafts [36]. It is conceivable that such microdomains act as concentration platforms for membrane-associated proteins, and hence may mediate quick oligomerization of pore forming toxins [35,37,38]. It has actually been shown that pore formation of *S. aureus* Hla is highly effective in biological membranes, which have a high proportion of SM [39]. However, it was not clear whether the lipid composition affects the binding of the monomers, the assembly of membrane-bound monomers to heptamers, or the final step of pore formation, namely the coordinated unfolding of the stem loops of each of the assembled monomers to form the transmembrane portion of the pore.

To answer these questions, we used the recombinant form of another toxin of *S. aureus*, hemolysin B (beta-hemolysin, Hlb), which has enzymatic activity and functions as a neutral sphingomyelinase [40,41] cleaving SM to phosphorylcholine and ceramide [42,43]. Immortalized human airway epithelial cells (S9, 16HBE14o-), as well as freshly isolated human nasal epithelium, were pre-treated with recombinant Hlb to deplete sphingomyelin from the PM and subsequently exposed to recombinant Hla. Cells were tested for the well-known effects of Hla on cell signaling like calcium influx [17], changes in cell morphology, and cell layer integrity [18], or actin remodeling induced by hypo-phosphorylation of cofilin at Ser3 [19].

## 2. Results

### 2.1. Pre-Treatment of Airway Cells with rHlb Allows rHla Monomer Binding to the PM, but Prevents Formation of Heptamers

To investigate whether SM is necessary for binding of Hla monomers to the host cell plasma membrane or assembly of heptameric transmembrane pores in these membranes, we pre-treated confluent cell layers (16HBE14o- and S9) for 1 h with 5000 ng/mL rHlb (sphingomyelinase) followed by a 0–4 h incubation with 2000 ng/mL rHla. Semi-quantitative Western blot analysis in whole cell protein extracts showed no changes in abundance of Hla monomers, whether cells had been pre-treated with rHlb or not (Figure 1A,B,D,E). This indicates that binding of rHla monomers to airway cell plasma membranes was obviously not affected by the removal of SM from the plasma membranes in the two cell types (c.f. Appendix A). Experiments using freshly prepared human nasal tissue showed similar results confirming the above conclusion (Figure 1G,H). In contrast, the abundance of rHla heptamers was lower in cell or tissue samples that were pre-treated with sphingomyelinase (rHlb) (Figure 1C,F,I). In 16HBE14o- as well as S9 cells that had not been pre-treated with rHlb, heptamer assembly started immediately after the addition of rHla at 0 h and increased steadily with the duration of exposure up to 4 h (Figure 1C,F). However, when cells had been pre-treated with rHlb, heptamer abundance was significantly lower and did not show any increases over the time of exposure (Figure 1C,F). Similar results were obtained when freshly prepared human nasal tissue was used in the experiments (Figure 1H,I). These observations indicate that the presence of SM in the plasma membranes of human airway epithelial cells is essential for *S. aureus* Hla to form multimeric complexes.

### 2.2. Effects of Sphingomyelinase Pre-Treatment of Airway Epithelial Cells on rHla-Mediated Changes in [Ca^2+^]_i_

As previously shown in human airway epithelial cells, treatment with rHla induced elevations in the cytosolic calcium concentration ([Ca^2+^]_i_) [15,17]. As observed previously, [Ca^2+^]_i_ started to increase with a lag phase of approximately 5–10 min after the addition of rHla and reached levels significantly different (*p* < 0.05) from the controls at 20–22 min recording time. These results were confirmed in this study as treatments of 16HBE14o- (Figure 2A), as well as S9 cells (Figure 2B), with 2000 ng/mL rHla resulted in significant increases in [Ca^2+^]_i_ (traces PBS + rHla). Pre-treatment of 16HBE14o- (Figure 2A) or S9 cells (Figure 2B) with 5000 ng/mL rHlb (sphingomyelinase) and subsequent exposure to 2000 ng/mL rHla (traces rHlb + rHla), however, did not result in any significant increases in [Ca^2+^]_i_. These traces were not significantly different from those that were obtained using cells that had been pre-treated with PBS (instead of rHlb) and treated with PBS instead of rHla during the experiment (Figure 2, traces PBS + PBS). Treatments of 16HBE14o- or S9 cells with 5000 ng/mL rHlb (traces rHlb + PBS), per se, did not induce any changes in [Ca^2+^]_i_ when compared to untreated control cells. These results indicate that pre-treatment of airway epithelial model cells with sphingomyelinase (rHlb) prevented rHla-mediated increases in [Ca^2+^]_i_. Because acute addition of sphingomyelinase (rHlb) to airway epithelial cells did not elicit any sustained changes in [Ca^2+^]_i_ (Appendix A) and rHlb-pre-treated cells showed strong calcium influx upon addition of calcium ionophores (Appendix A), it can be concluded that the suppression of rHla-mediated calcium signaling in rHlb-pretreated airway epithelial cells is not a consequence of indirect effects (like rHlb-mediated emptying of calcium stores before the addition of rHla or ceramide-mediated internalization of rHla-containing plasma membrane).

### 2.3. Effects of Sphingomyelinase Pre-Treatment of Airway Epithelial Cells on rHla-Mediated Formation of Paracellular Gaps

Previous investigations had demonstrated alterations in cell shape, loss of cell-cell contacts, and the formation of paracellular gaps in initially confluent cell layers of airway epithelial model cells upon exposure to rHla [18], with 16HBE14o- cells’ reaction being much more pronounced than S9 cells. Thus, we tested in this study whether these effects of rHla could be moderated or abrogated by pre-treatment of cells with sphingomyelinase (rHlb). As shown in the still pictures taken from time lapse movies shown in Figure 3 (third row, each), confluent cell layers of 16HBE14o-, as well as S9 cells that had been pre-treated with 5000 ng/mL rHlb, did not develop microscopically visible gaps or other rHla-typical cellular changes upon treating the cells with 2000 ng/mL rHla (added at 0 h). Cell growth, division, and shape were comparable with control cells treated with PBS in both cell lines (Figure 3, first row, each), but clearly different from those cell cultures that had not been pre-treated with sphingomyelinase and exposed to rHla (Figure 3, second row, each).

### 2.4. Effects of Pre-Treatment of Airway Epithelial Cells with Sphingomyelinase (rHlb) on rHla-Mediated Hypo-Phosphorylation of Cofilin

Earlier studies have shown that rHla-treatment of airway epithelial model cells resulted in hypo-phosphorylation of the actin depolymerizing factor cofilin [19], which is likely to be the most important part in the chain of events leading to rHla-mediated changes in cell shape and paracellular gap formation. In this study, we tested the impact of rHla on pSer3-phosphorylation of cofilin with or without pre-treatment of cells with rHlb. As reported previously [19], and again shown in Figure 4, treatment of human airway epithelial cells (16HBE14o- and S9 cells) with 2000 ng/mL rHla significantly decreased the levels of pSer3-cofilin in 16 HBE14o- cells, as well as in S9 cells (Figure 4B,D, dots). After adding rHla to the 16HBE14o- cell culture, the level of pSer3-cofilin declined, reaching a minimum after 2 h that was maintained for the remaining experimental time (Figure 4B). In S9 cells, the decrease of cofilin phosphorylation was transient and started to recover between 2 and 4 h of rHla-exposure (Figure 4D). These results are in accordance with those reported previously [19]. When cells had been pre-incubated with 5000 ng/mL *S. aureus* rHlb, no decline in cofilin phosphorylation was observed in either cell type (Figure 4B,D, diamond). Total cofilin abundance (normalized to β-actin), although somewhat variable in cells, was not significantly affected by any of these treatments.

When freshly prepared human nasal tissue was exposed to 2000 ng/mL rHla for 2 h, a significant decline in pSer3-phosphoration of cofilin was observed compared with untreated control tissue (Figure 4F). However, when nasal tissue was pre-incubated with 5000 ng/mL rHlb, rHla-mediated hypo-phosphorylation of cofilin was absent (Figure 4F). This indicates that the results obtained using 16HBE14o- or S9 airway model cells are of physiological relevance.

## 3. Discussion

Besides sphingomyelin (SM), the most common and important phospholipids in the plasma membrane of eukaryotic cells are phosphatidylcholine (PC), phosphatidylserine (PS), and phosphatidylethanolamine (PEA) [44]. The choline containing lipids, SM and PC, have been implicated in the formation of lipid rafts [38,45,46]. SM has been discussed as an important factor mediating the deleterious effects of *S. aureus* alpha-toxin (Hla) [21,47] on host cells [35,39]. However, it was not clear which of the sequential step(s) of forming functional transmembrane pores in the plasma membranes of host cells (monomer binding, monomer heptamerization, and pre-pore assembly or unfolding of the stem loops of each of the monomers to from the functional transmembrane pore) were facilitated by sphingomyelin. We used *S. aureus* β-hemolysin (Hlb), another secreted virulence factor of *S. aureus* that functions as a neutral sphingomyelinase [42,43] (Appendix A) as a tool to deplete the pool of sphingomyelin in the outer leaflet of the plasma membranes of airway epithelial model cells.

To investigate whether SM is necessary for binding Hla monomers to the plasma membrane or formation of heptameric pores in the plasma membrane, immortalized human airway epithelial cells (16HBE14o-, S9), as well as freshly isolated human nasal tissue, were pre-treated with Hlb and subsequently exposed to rHla. As shown in Figure 1, the removal of SM (rHlb + rHla) had no effect on the abundance of Hla monomers in the protein extracts from cells when compared with the samples obtained from cells that had not been pre-treated with Hlb (Figure 1B,E,H). This indicates that Hla monomer binding was not affected by the presence or absence of SM. Western blot signals associated with Hla heptamers, however, showed a completely different picture (Figure 1C,F,I). Because heptamer formation by membrane-bound alpha-toxin monomers is a very rapid process [48], the abundance of Hla heptamers in rHla-treated airway epithelial model cells steadily increased over the time of Hla exposure in those cells that were not pre-treated with Hlb (Figure 1C,F; rHla). In cells that had been pre-treated with sphingomyelinase (rHlb), however, the abundance of Hla heptamers was significantly lower or entirely absent and did not change over time during the incubation period (Figure 1C,F; rHlb + rHla). The latter finding was confirmed when freshly isolated human airway tissue was used (Figure 1I), indicating that the lack of Hla heptamer formation is a general feature of SM-depleted airway epithelial cells. These results indicate that the presence of SM in plasma membranes of airway epithelial cells is dispensable for the attachment of toxin monomers to the cell surface, but essential for the heptamerization process and the formation of the pre-pore. Without the formation of a pre-pore there should be no unfolding of the stem loops of the seven subunits and no formation of a functional transmembrane pore. Thus, our expectation was that none of the usual Hla-mediated cell physiological changes would occur upon Hla exposure of our eukaryotic model cells if these had been pre-treated with sphingomyelinase.

In the absence of SM in airway epithelial cell membranes, rHla heptamerization was almost completely suppressed. The residual multimer formation may be due to incomplete SM degradation during the pre-incubation of cells with rHlb. Alternatively, choline-containing lipids like PC may be able to replace SM to a certain extent in mediating Hla heptamer formation [35]. Another potential explanation would be that Hla monomers assemble spontaneously and at a low rate without any assistance from lipids. This conclusion is supported by the observation that rHla monomers maintained for 10 min in aqueous solution at room temperature (Figure 1, pos con) sometimes show spontaneous heptamerization without the need for any additional reagents.

While these options remain to be tested, we focused on the question whether the removal of SM from host cell membranes and almost complete suppression of heptamer formation may be able to suppress the cell physiological effects that are normally seen in Hla-treated cells upon pore-formation [15,18,19]. We chose to measure the time course of changes in [Ca^2+^]_i_, the formation of paracellular gaps, as well as the hypo-phosphorylation of cofilin upon addition of rHla to airway epithelial model cells that had or had not been pre-treated with rHlb.

Exposure of Indo-1-loaded 16HBE14o- cells or S9 cells to rHla resulted in a slow increase in Ca^2+^-mediated dye fluorescence, whose onset was delayed for 3 to 8 min, probably due to the time required for generating Hla pore-mediated calcium influx that exceeded the capacity of endogenous calcium extrusion mechanisms in these cells (Figure 2). These results match those that have been reported previously [15,18]. Pre-treatment of cells with sphingomyelinase (rHlb), however, completely abolished this cellular response to rHla-exposure (Figure 2, traces rHlb + rHla), indicating that sphingomyelin depletion prevents Hla from forming functional transmembrane pores. The few residual heptamers that seemed to form in rHlb-treated cells (Figure 1C,F) may allow some influx of calcium ions into the cytosol of these cells that is, however, not large enough to out-perform the endogenous calcium extrusion mechanisms.

Monitoring of cell shape changes and paracellular gap formation in confluent cultures of 16HBE14o-, as well as S9 cells during exposure to rHla using time lapse microscopy, we could confirm that treatment of cells with rHla induced loosening of the cells from each other and from the culture dish (Figure 3, traces rHla). These effects were much more pronounced in 16HBE14o- cells than in S9 cells, a finding that confirmed the results of previous studies [18,19]. However, when cells had been pre-treated with sphingomyelinase (rHlb), these effects were completely suppressed, and the cultures maintained their confluent appearance over the entire experimental period despite the continued presence of rHla (Figure 3, traces rHlb + rHla).

The rHla-mediated changes in shell shape and paracellular gap formation are associated with and most likely caused by the disruption of the original architecture of the actin cytoskeleton in rHla-treated airway epithelial model cells [19]. We monitored the level of pSer3-phosphorylation of the actin depolymerizing factor cofilin that has been shown to be downregulated upon rHla exposure of cells [19]. As shown in Figure 4 (traces with dots), we could confirm the previous findings that cofilin hypo-phosphorylation occurred under rHla-treatment in airway epithelial model cells with a sustained effect in 16HBE14o- and a transient effect in S9 cells. However, when cells had been pre-treated with rHlb before the start of the rHla-experiment there was no indication of any change in pSer3-phosphorylation in cofilin over the experimental period of 4 h (Figure 4, traces with diamonds). Parallel experiments using freshly isolated human nasal tissue gave similar results (Figure 4F). This indicates that pre-treatment of airway epithelial cells with sphingomyelinase (rHlb) prevents the activation of signaling pathways and any of the detrimental cell physiological effects usually induced by rHla exposure.

An interesting question was whether the attenuation of Hla-mediated cell damage by sphingomyelinase pre-treatment is a phenomenon limited to airway epithelial cells, or a more general effect in eukaryotic cells. To test this we used sheep erythrocyte agar plates and pre-treated the red blood cells in the agar matrix with sphingomyelinase (rHlb) before exposing them to rHla. Exposing cells to sphingomyelinase (rHlb) results in some changes in cell integrity (Appendix A), known as “incomplete hemolysis” [9]. As shown in Appendix A, these cells did not show the typical complete hemolysis that occurs in erythrocytes exposed to rHla (Appendix A). We concluded that pre-treatment of sheep erythrocytes with sphingomyelinase (*S. aureus* rHlb) suppresses rHla-mediated hemolysis. This indicates that the removal of sphingomyelin from the outer leaflet of the plasma membranes of eukaryotic cells may generally save eukaryotic cells from the deleterious actions of *S. aureus* alpha-toxin. Such a conclusion is consistent with the recent observation that cells lacking sphingomyelin synthase 1 (SGMS1) are resistant against Hla-virulence [25].

## 4. Materials and Methods

### 4.1. Chemicals and Reagents

Indo-1/AM was obtained from Invitrogen (Karlsruhe, Germany). Trypsin (with EDTA) was purchased from PAN-Biotech (Aidenbach, Germany). Lipid standards were ordered from Sigma (Steinheim, Germany). Antibodies (Ab) were obtained from these sources: Hla-Ab (S7531) and β-Actin-Ab (AC-15, A5441) from Sigma (Steinheim, Germany), p-Cofilin1-Ab (hSer3-R, sc-12912-R), cofilin-Ab (E-8, sc-376476), goat anti-rabbit IgG-HRP (sc-2004) and goat anti-mouse IgG-HRP (sc-2005) from Santa Cruz (Heidelberg, Germany). All other chemicals were reagent grade and obtained from Roth (Karlsruhe, Germany). 

### 4.2. Expression and Purification of Recombinant Staphylococcus aureus Hla and Hlb

Recombinant alpha-toxin (rHla) and recombinant beta-toxin (rHlb) were prepared and purified as described previously [49]. The purity of rHla and rHlb was assessed by SDS-PAGE. The concentration of rHla routinely used was 2000 ng/mL (60 nmol/L), for reasons discussed previously [18]. Exposure of airway epithelial cells to such a concentration of rHla for 2 h results in cell death of less than 15% of all cells [17]. Lower concentrations of rHla may induce pore formation as well, but the physiological responses are much less pronounced and can hardly be detected. As we have previously shown, treatment of cells with 200 ng/mL rHla does not induce MAP kinase activation [49] or intracellular calcium accumulation [18].

### 4.3. Cell Culture

Immortalized human airway epithelial cells (16HBE14o- or S9) were obtained from Karl Kunzelmann (Regensburg, Germany, 16HBE14o-) or from ATCCLGC Standards (Wesel, Germany, S9) and cultured on 10 cm Cell+ plates (Sarstedt, Numbrecht, Germany) at 37 °C and gassed with 5% CO_2_. Eagle’s MEM (PAN-Biotech, Aidenbach, Germany) containing 10% FBS superior 168 (Biochrom, Berlin, Germany) and 1% (w/v) penicillin/streptomycin (PAA 169 Laboratories, Cölbe, Germany) was routinely used as cell culture medium and changed every 3–4 days. Shortly before cells formed confluent monolayers, they were passaged on new cell culture plates or directly used in the experiments. All cell cultures were checked for *Mycoplasma* contamination on a regular basis.

### 4.4. Freshly Prepared Human Airway Tissue

Primary human airway tissue was isolated from the ethmoid sinus uncinate process in chronic rhinosinusitis patients with macroscopically normal mucosa undergoing surgery for the removal of nasal polyps. The sheet of epithelial and underlying connective tissue was lifted from the bone, rinsed several times in cell culture medium (see above), and transferred to the lab. Tissue was sliced in a vertical direction (0.5 mm thickness) to maintain the original tissue structure. The experiments were approved by the ethics committee of the university hospitals in Greifswald (BB95/10; September 1, 2010) and Münster (2017-120-f-S). Informed written consent was obtained from all donors.

### 4.5. Sample Preparation for Western Blotting

Immortalized human airway epithelial cells (16HBE14o- or S9) were incubated in the presence or absence of 5000 ng/mL recombinant *S. aureus* Hlb (sphingomyelinase) for 1 h. Afterwards, cells were treated with 2000 ng/mL recombinant Hla for 0, 1, 2, or 4 h or with phosphate buffered saline (PBS) (control). Primary human airway tissue was treated with 5000 ng/mL rHlb for 1 h and subsequently with 2000 ng/mL rHla for 2 h or with PBS (control). After incubation of cells or tissue in the presence or absence of rHla, the culture medium was carefully aspirated and the material was washed using 5 mL PBS. To each 10 cm-plate of cultured cells, 400 μL lysis buffer (100 mmol/L KCl, 20 mmol/L NaCl, 2 mmol/L MgCl_2_, 0.96 mmol/L NaH_2_PO_4_, 0.84 mmol/L CaCl_2_, 1 mmol/L EGTA, 0.5% (*v*/*v*) Tween 20, 25 mmol/L HEPES (free acid), pH 7.2 containing 10 mmol/L each of aprotinin, leupeptin, and pepstatin, as well as 100 mmol/L PMSF and 0.33 mmol/L ortho-vanadate) was added. Cells were scraped off the cell culture plate using a cell scraper. The suspension of cytosolic extract and particulate matter was transferred to a 1.5 mL Eppendorf-reaction tube and immediately transferred on ice. The samples were homogenized on ice using a T8-Ultraturrax (IKA Labortechnik, Staufen, Germany) for 30 s, each, and combined with an equal volume of SDS sample-buffer, mixed, and frozen at −80 °C [19].

### 4.6. Semi-Quantitative Western Blotting

Proteins were separated by SDS/PAGE (10% or 13% gels) in a minigel apparatus (BioRad, Munich, Germany) and transferred to nitrocellulose membrane (HP40, Roth, Karlsruhe, Germany) [19]. Western blotting for the quantification of total or phosphorylated proteins was performed using (phospho-) specific antibodies, HRP-linked secondary antibodies (1:6000), and enhanced luminescence reagents (Biozym, Oldendorf, Germany). Signals were recorded using a Fusion FX7-SL gel imager (Vilbert Lourmat, Eberhardzell, Germany). Band signal intensities were assessed by densitometry using Phoretix 1 D (Nonlinear Dynamics, Newcastle upon Tyne, UK). To correct for potential minor differences in exposure time, the mean density of all bands on each gel image was used to normalize the densities of individual bands of the same gel. Signal intensities of the phosphorylated forms of proteins were normalized to the signals obtained using antibodies against the respective core proteins or against the β-actin band densities. Relative band densities were used to calculate means and standard deviations of different experiments.

### 4.7. Intracellular Calcium Concentrations

Changes in intracellular calcium concentration ([Ca^2+^]_i_) were monitored in airway epithelial cells using the calcium sensitive indicator dye indo-1, as described previously [15,18]. The cell suspension was split in equal aliquots after the dye loading procedure. Cells in one portion were treated with 5000 ng/mL rHlb, and cells in the other portion with phosphate buffered saline (PBS) during the 30 min recovery period. Subsequently, cells were washed 3 times (2 min at 600× g, each) and finally resuspended in 1.5 mL HEPES-buffered saline. Portions of 300 µl cell suspension were transferred to each well of a 96-well-plate (black flat-bottomed microplate, Biozym, Oldendorf, Germany) and treated with 2000 ng/mL rHla or PBS as control after a pre-run of 12 min. Calcium concentrations in the samples were determined using the Infinite M200Pro microplate reader (Tecan, Crailsheim, Germany) equipped with the software package I-control V1.11 (Tecan, Grödig, Austria, 2014) at a constant temperature (37 °C). Excitation wavelength was set to 338 nm with a slit width of 9 nm, emission wavelength was set to 400 nm with a slit width of 20 nm. Fluorescence data were recorded in intervals of 12 s. All fluorescence intensity values during the individual measurements were normalized to the average fluorescence intensities during the initial 6 min measuring period and expressed in % of these pre-run intensities.

### 4.8. Time Lapse Microscopy

Airway epithelial cells (16HBE14o-, S9) were cultured in 35 mm μ-cell culture plates (Ibidi, Planegg, Germany) in medium as described above until they reached confluence. The medium was renewed one day before the plate was transferred to the time lapse-microscope (Biostation II, Nikon Instruments, Düsseldorf, Germany). The microscope chamber was thermostatically controlled at 37 °C and gassed with 5% CO_2_ in air during the experiment. Images of cells were taken every 3 min over 24 h and combined to time lapse-movies of 30 s duration.

### 4.9. Data Presentation and Statistics

Data are presented as means and S.D. of n experiments on different cell/tissue preparations. Significant differences in the series of means were detected by ANOVA. Individual means were tested for significant differences to the appropriate controls using Student’s *t*-test (used if variances were equal) or Welch’s *t*-test (w, used if variances were not equal). Significant differences of means were assumed at *p* < 0.05.

## Figures and Tables

**Figure 1 toxins-11-00126-f001:**
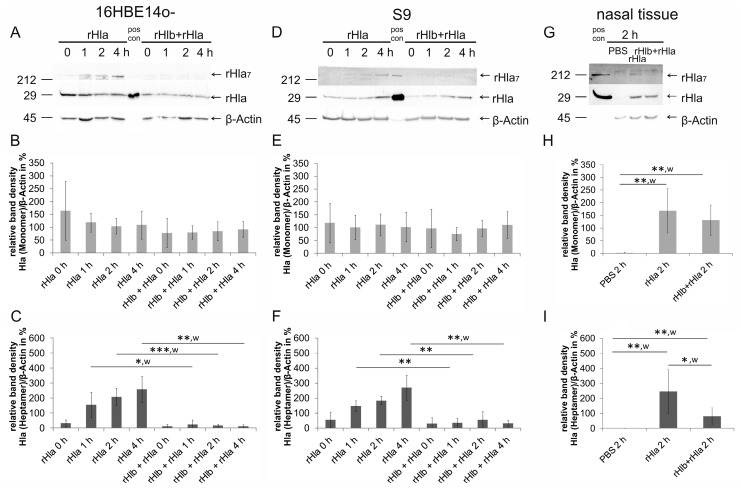
Pre-incubation of cells with sphingomyelinase (rHlb) prevented formation of rHla heptamers (rHla_7_), but not plasma membrane binding of rHla monomers in airway epithelial cells and nasal tissue. Confluent layers of immortalized airway epithelial cells (16HBE14o- (**A**–**C**) and S9 (**D**–**F**)) were treated with 2000 ng/mL rHla after pre-treatment of cells in the presence or absence of 5000 ng/mL rHlb (sphingomyelinase) for 0–4 h. Cells treated with rHla showed binding of Hla monomers (33 kDa, rHla) and Hla heptamers (231 kDa, rHla_7_). The rHla monomer abundances were independent of the incubation time with rHla und independent of the pre-treatment regime with sphingomyelinase (**A**,**B**,**D**,**E**). Formation of Hla heptamers, however, was significantly reduced in 16HBE14o- or S9 cells which had been pre-treated with sphingomyelinase (rHlb) compared with control cells without sphingomyelinase pre-treatment (**A**,**C**,**F**). Experiments using freshly prepared human nasal tissue showed similar results (**G**–**I**). Representative example Western blot signals of Hla heptamers (rHla_7_), Hla monomers (rHla), and β-actin are shown (**A**,**D**,**G**). Recombinant Hla (approximately 40 ng/lane) was used to indicate the position of Hla monomers (pos con), and in some cases, heptamers that form spontaneously when aqueous solutions of rHla are left at room temperature for 10 min. The positions of molecular mass standards (in kDa) are indicated. Mean values ± S.D. of densitometry signals of Western blot analyses normalized to the densities of the respective β-actin bands (*n* = 5, each) were assembled in histograms. Individual means were tested for significant differences using Student’s *t*-test or Welch’s *t*-test (*w*): * *p* < 0.05, ** *p* < 0.01, or *** *p* < 0.001.

**Figure 2 toxins-11-00126-f002:**
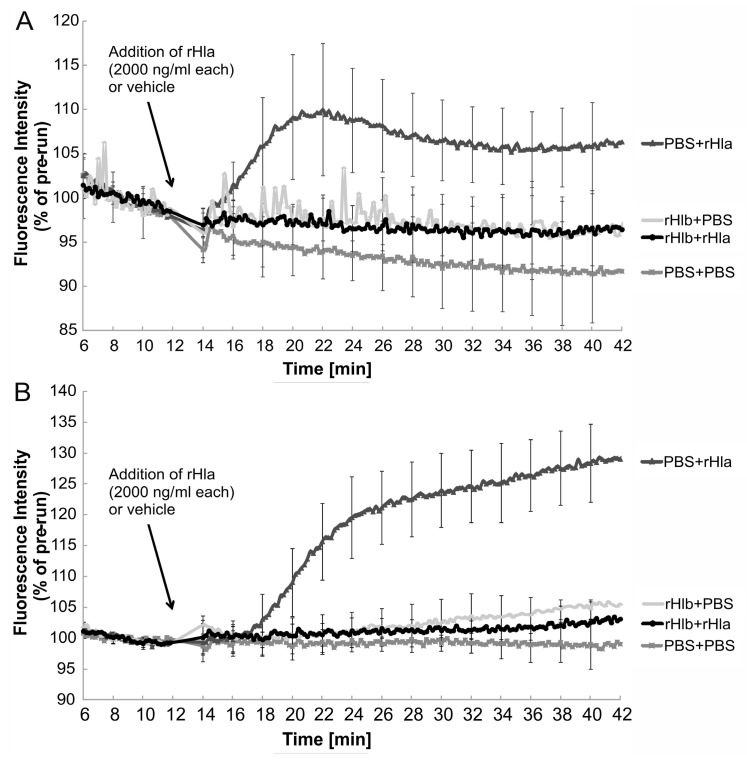
Pre-incubation of cells with sphingomyelinase (rHlb) abrogated rHla-mediated increases in [Ca^2+^]_i_ in airway epithelial cells. Calcium-dependent fluorescence intensities (normalized to the mean fluorescence level during the initial 6 min of each recording) were monitored in Indo-1 loaded 16HBE14o- (**A**) or S9 cells (**B**), respectively. Cells were pre-treated in the absence or presence of rHlb as controls (PBS + PBS; rHlb + PBS) or additionally treated with 2000 ng/mL rHla (PBS + rHla; rHlb + rHla) during the experiment. Pre-incubation of cells with rHlb prevented rHla-mediated increases in [Ca^2+^]_i_ (rHlb + rHla, circles) compared with those cells that were not pre-exposed to rHlb (PBS + rHla, triangles). Mean traces (± S.D. at 2 min intervals) of *n* = 9 (16HBE14o-, **A**) or *n* = 5 (S9, **B**) independent cells preparations are shown.

**Figure 3 toxins-11-00126-f003:**
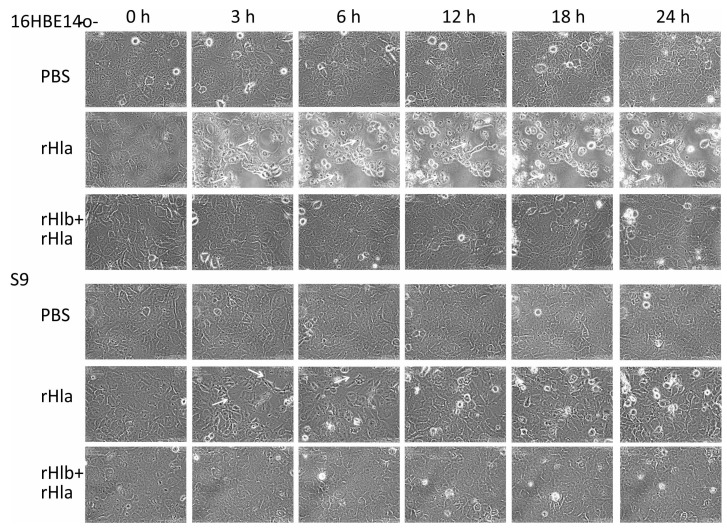
Pre-incubation of cells with sphingomyelinase (rHlb) prevented rHla-mediated formation of microscopically visible gaps in airway epithelial cell layers. Cellular changes, like loss of contacts to neighboring cells, changes in cell shape, and formation of paracellular gaps (white arrows) induced by *S. aureus* alpha-toxin (rHla, added at 0 h) in initially confluent cell layers of human airway epithelial model cells (16HBE14o- or S9 cells) were inhibited by pre-incubation of cells with *S. aureus* beta-toxin (rHlb), a sphingomyelinase. Treatments of cells with rHlb alone showed no cellular changes (data not shown). Phase contrast images were taken at different time points after adding PBS (vehicle control, PBS), 2000 ng/mL rHla, or rHla in the continued presence of 5000 ng/mL rHlb, respectively. Images were taken at the indicated times from time-lapse movies (Biostation IM, Nikon) monitoring the cells over 24 h.

**Figure 4 toxins-11-00126-f004:**
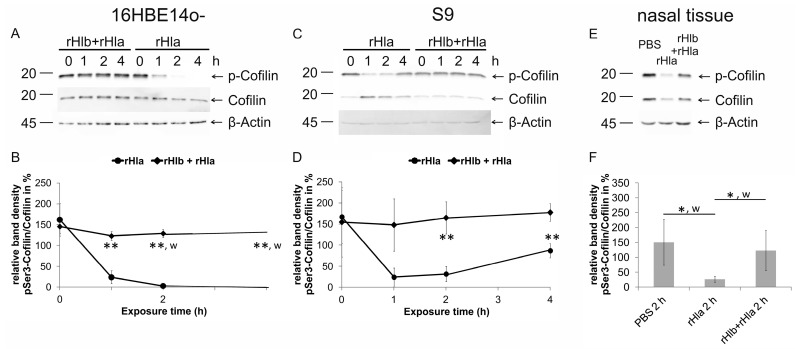
Pre-incubation of cells with sphingomyelinase (rHlb) prevented rHla-mediated hypo-phosphorylation of cofilin in airway epithelial cells and nasal tissue. Ser3-cofilin phosphorylation was monitored in whole cell extracts of 16HBE14o- cells (**A**,**B**) and S9 cells (**C**,**D**) that had been pre-treated (1 h) with 5000 ng/mL rHlb (rHlb) or not (PBS, rHla) and subsequently treated with rHla (rHla, 2000 ng/mL) either in the absence or continued presence of 5000 ng/mL rHlb by semi-quantitative Western blotting. In addition, primary human nasal tissue samples (**E**,**F**) were pre-treated (1 h) with 5000 ng/mL rHlb (rHlb) or not (PBS, rHla) and subsequently treated with PBS (vehicle control) or 2000 ng/mL rHla (rHla), either in the absence or continued presence of 5000 ng/mL rHlb for 2 h. Representative pSer3-cofilin-, cofilin, and β-actin Western blot signals are shown (**A**,**C**,**E**: positions of molecular mass standards in kDa are indicated). Means ± S.D. of *n* = 3 (**B**,**D**) or *n* = 5 (**F**) independent preparations. Individual means were tested for significant differences using Student’s *t*-test or Welch’s *t*-test (*w*): * *p* < 0.05, ** *p* <0.01.

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
