# Peer review of "Sphingomyelin Depletion from Plasma Membranes of Human Airway Epithelial Cells Completely Abrogates the Deleterious Actions of S. aureus Alpha-Toxin"

_toxins, 2019, doi:10.3390/toxins11020126_

Round 1

Reviewer 1 Report

In the manuscript, the authors studied the importance of sphingomyelin (SM) for the actions of Staphylococcus aureus alpha-toxin to the airway epithelial cells or nasal tissue.

The results shown indicate that, by the pre-treatment of the cell membrane with sphingomyelinase, the toxin-binding to the cell membrane was not affected, but the toxin-oligomerization and the toxin-actions causing disturbance of the cell functions were strongly suppressed.

Namely, SM is demonstrated to be essential for the actions of the alpha-toxin after binding to the cell membrane.

These findings are thought to be useful for researchers in the similar fields.

However, some minor modifications are required as follows:

1. Line 48: calcium ion is not a monovalent cation.

2. Line 70, 76 and 208: sphingomyelin or sphingomyelin (SM) is SM.

3. Line 76, 209 and 245: phosphatidylcholine or phosphatidylcholine (PC) is PC.

4. Figure 1G: rHlb is rHlb + rHla.

5. line 230-231: (Figure 2C, F; rHla) is (Figure 1C, F; rHla).

Author Response

In the manuscript, the authors studied the importance of sphingomyelin (SM) for the actions of Staphylococcus aureus alpha-toxin to the airway epithelial cells or nasal tissue.

 The results shown indicate that, by the pre-treatment of the cell membrane with sphingomyelinase, the toxin-binding to the cell membrane was not affected, but the toxin-oligomerization and the toxin-actions causing disturbance of the cell functions were strongly suppressed.

Namely, SM is demonstrated to be essential for the actions of the alpha-toxin after binding to the cell membrane.

These findings are thought to be useful for researchers in the similar fields.

However, some minor modifications are required as follows:

1. Line 48: calcium ion is not a monovalent cation.

Response: We are thankful to the reviewer for this constructive comment! We have changed the wording.

2. Line 70, 76 and 208: sphingomyelin or sphingomyelin (SM) is SM.

Response: Has been corrected.

3. Line 76, 209 and 245: phosphatidylcholine or phosphatidylcholine (PC) is PC.

Response: Has been corrected.

4. Figure 1G: rHlb is rHlb + rHla.

Response: We are thankful to the reviewer for this hint! We have corrected the heading.

5. line 230-231: (Figure 2C, F; rHla) is (Figure 1C, F; rHla).

Response: We are thankful to the reviewer for this constructive comment! We have corrected this mistake.

Reviewer 2 Report

Summary:

This is an interesting study probing the possibility that sphingolipids are required to assemble Hla monomers into functional heptameric complexes. While the phenotypes are interesting, the authors have treated the lung epithelial cells in these experiments as passive participants, assigning all observed phenotypes to Hla without accounting for possible contributions driven by Hlb-alterations of epithelial cell biology. Several controls are required to determine whether the observed phenotypes are due to altered Hla assembly/function rather than Hlb-driven alterations to epithelial cell biology.

Major Points:

1.       Some important controls are missing from the rHlb depletion experiments that are required to account for possible alternative mechanisms driving some of the observed phenotypes. The rHlb treatment depletes all of (or nearly all of) the cellular sphingomyelin; representing a depletion of roughly 1/3rd of the plasma membrane lipid composition. This is a massive alteration in the structure/composition of the membrane whose potential effects on the epithelial cells are completely ignored in this study. Endocytosis is profoundly affected by lipid depletion, and sphingomyelin is known to be important for multiple endocytic pathways. Alterations to the bulk uptake of Hla monomers due to alterations in endocytosis could easily be mistaken for the apparent uniform “binding” of monomeric Hla in Figure 1. Likewise, Ca2+ flux is treated as if it could only be generated by Hla heptamer formation – completely ignoring that alpha toxin (in S. aureus and numerous other species) has been shown to interact with and alter the activity of epithelially-expressed calcium channels (e.g. Eichstaedt  et al, 2008). A number of controls are required to assess these possibilities:

a.       The rate of endocytosis should be assessed with/without rHlb addition. If the investigators have access to a fluorescent microscope or flow cytometer a simple way to quantify this is to measure to uptake of fluorescently-labeled dextran (at least 70 kDa in mass), which is taken up non-specifically from the fluid phase during endocytosis.

b.       The binding and calcium-flux experiments should be repeated in the presence of a pan-inhibitor of endocytosis (e.g. N-ethylmaleimide)

c.       A positive control for calcium flux is required, to show that rHlb is not affecting the cells intrinsic ability to engage in a Ca2+ flux. PMA stimulation may be sufficient.

2.       Figure 3 does not support the author’s conclusion that there was no formation of paracellular gaps, as without a change in cellular adhesion or cell shape, loss of cell-cell junctions are not apparent in white-light images. A more rigorous assay of paracellular gap formation (e.g. transwell diffusion assay) need to be performed before the authors can make this conclusion.

3.       How was the concentration of rHlb and rHla used in experiments determined? Is there a dose-dependent inhibition of rHla activity when the concentration of rHlb is changed?

Minor Points:

1.       There are minor typos throughout the document – e.g. on line 61 there is an unnecessary opening bracket “(“ before alpha5-beta1. Please correct these.

2.       Line 230-231 refers to Figure 2C,F – these figure panels do not exist. I suspect the authors intended to reference Figure 1 here.

3.       The results of Figure S2 are not convincing, as a higher diffusivity of Hlb vs Hla, or even differences in catalytic rate, would produce a similar result.

Author Response

This is an interesting study probing the possibility that sphingolipids are required to assemble Hla monomers into functional heptameric complexes. While the phenotypes are interesting, the authors have treated the lung epithelial cells in these experiments as passive participants, assigning all observed phenotypes to Hla without accounting for possible contributions driven by Hlb-alterations of epithelial cell biology. Several controls are required to determine whether the observed phenotypes are due to altered Hla assembly/function rather than Hlb-driven alterations to epithelial cell biology.

Major Points:

1.       Some important controls are missing from the rHlb depletion experiments that are required to account for possible alternative mechanisms driving some of the observed phenotypes. The rHlb treatment depletes all of (or nearly all of) the cellular sphingomyelin; representing a depletion of roughly 1/3rd of the plasma membrane lipid composition. This is a massive alteration in the structure/composition of the membrane whose potential effects on the epithelial cells are completely ignored in this study. Endocytosis is profoundly affected by lipid depletion, and sphingomyelin is known to be important for multiple endocytic pathways. Alterations to the bulk uptake of Hla monomers due to alterations in endocytosis could easily be mistaken for the apparent uniform “binding” of monomeric Hla in Figure 1. Likewise, Ca2+ flux is treated as if it could only be generated by Hla heptamer formation – completely ignoring that alpha toxin (in S. aureus and numerous other species) has been shown to interact with and alter the activity of epithelially-expressed calcium channels (e.g. Eichstaedt  et al, 2008). A number of controls are required to assess these possibilities:

a.       The rate of endocytosis should be assessed with/without rHlb addition. If the investigators have access to a fluorescent microscope or flow cytometer a simple way to quantify this is to measure to uptake of fluorescently-labeled dextran (at least 70 kDa in mass), which is taken up non-specifically from the fluid phase during endocytosis.

b.       The binding and calcium-flux experiments should be repeated in the presence of a pan-inhibitor of endocytosis (e.g. N-ethylmaleimide)

c.       A positive control for calcium flux is required, to show that rHlb is not affecting the cells intrinsic ability to engage in a Ca2+ flux. PMA stimulation may be sufficient.

Response: SMase treatment of cells does actually not "deplete" the plasma membrane from one third of the plasma membrane lipids, but just removes the polar head group of SM from the remaining ceramide moiety (c.f. Chatterjee, S. (1999) Chem Phys Lipids 102(1): 79-96). It is true that some ceramides may indeed leave the membrane and have signaling effects on their own, but these may affect mainly other cell types that are not present in our system as the overall survival rate of the airway epithelial cells is not affected by rHlb treatment.

The reviewer's notion that the uniform abundance of Hla monomers in cell extracts may be caused by a bulk uptake of Hla monomers via (ceramide-induced?) endocytosis (c.f. Trajkovic, K. et al. (2008) Science 319(5867): 1244-1247) can be ruled out. As described in the methods part, we performed our analyses on whole cell protein extracts. A more rapid uptake of Hla monomers from the extracellular space or from the plasma membrane by endocytosis in rHlb pre-treated cells would result in an overall increase in Hla monomer abundance in such a sample compared with that of non-pre-treated cells. The results clearly indicate that such an effect is definitely not observed.

We have recently published a paper (Eiffler et al. (2016) American Journal of Physiology - Lung Cellular and Molecular Physiology 311: L676-L685) in which we show that rHla-mediated influx of sodium and calcium ions as well as plasma membrane potential effects were absent when cells were treated with the pore-blocker IB201 simultaneously with rHla or with the non-pore forming Hla-variant rHla-H35L. This shows that simple attachment of rHla monomers is not sufficient for the induction of Ca2+ flux. Only the formation of functional transmembrane pores allow the influx of Ca2+. We have added a graph in the supplementary material (Figure S2) showing that acute addition of rHlb to Indo-1 loaded cells did not induce any changes in [Ca2+]i in the long run indicating that indirect effects of rHlb pre-treatment (e.g. premature emptying of intracellular Ca2+ stores does not seem to occur). We have made this clear by adding a sentence in the results section.

2.       Figure 3 does not support the author’s conclusion that there was no formation of paracellular gaps, as without a change in cellular adhesion or cell shape, loss of cell-cell junctions are not apparent in white-light images. A more rigorous assay of paracellular gap formation (e.g. transwell diffusion assay) need to be performed before the authors can make this conclusion.

Response: For our purpose it was sufficient to show microscopically visible differences between cells that had been or had not been pre-incubated with rHlb. Our results definitely showed that there was no gap formation visible after pre-incubation of cells with Hlb and subsequent incubation with Hla. To avoid any confusion and to differentiate our results from those of TEER measurements, we have changed the wording in the text ("....microscopically visible ...").

3.       How was the concentration of rHlb and rHla used in experiments determined? Is there a dose-dependent inhibition of rHla activity when the concentration of rHlb is changed?

Response: Because Hlb has enzymatic activity, a dose-response-relationship is not the most meaningful thing to do. We performed preliminary experiments to determine the optimal conditions to apparently (visual inspection of stained HPTLC plates) remove the entire complement of sphingomyelin from the plasma membranes of experimental cells. For Hla, information on the rationale to use 2000 ng/ml has been given in Ref. 18 cited in the text (Materials and Methods, Chapter 4.2).

Minor Points:

1.        There are minor typos throughout the document – e.g. on line 61 there is an unnecessary opening bracket “(“ before alpha5-beta1. Please correct these.

Response: Has been corrected.

2.        Line 230-231 refers to Figure 2C,F – these figure panels do not exist. I suspect the authors intended to reference Figure 1 here.

Response: We are thankful to the reviewer for this constructive comment! We have corrected that mistake.

3.       The results of Figure S2 are not convincing, as a higher diffusivity of Hlb vs Hla, or even differences in catalytic rate, would produce a similar result.

Response:

The concern however, that differences in diffusibility of rHla and rHlb may be the reason for the observations does not appear valid to us as the toxins were washed into the gel matrix by applying small fluid volumes in a repetitive manner. That way, the fluid volume added to each well chiefly determines diffusion distance, not the geometry of the rHla or the rHlb molecules themselves. As the total volume of fluid added to one well is twice as high in the wells treated with combinations of rHlb and rHla (B, C) than in the others the diameter of the hemolysis area is slightly larger in B and C than that in D.

We test every batch of recombinant toxins routinely on sheep erythrocyte agar plates to check for proper biological activity of each preparation. Hemolysis of red blood cells is the 'classical' effect of Hla. Our experiment with combinations of Hlb and Hla was included in the supplement because we found this result interesting and indicative of the possibility that Hlb-mediated sphingomyelin removal may also prevent Hla-effects on other than airway epithelial cells. It was not meant as definitive scientific proof.

Reviewer 3 Report

The authors are presenting data that shows depletion of sphingomyelin by S. aureus beta hemolysin (a sphingomyelinase) prevents the toxic activity of alpha hemolysin. The authors propose that sphingomyelin is involved in oligomerization of Hla on the surface fo target cells.  The basic data supporting this notion is ralitively weak.  Data shown in FIgure 1 are the basis for this conclusion which is the primary novelty of the paper.  however the data shown are of poor quality.  The heptamer band is poorly visible.  to make this central point of the paper more convincing the authors need to provide data of higher quality.  

Also in Fig. 1G/H/I the authors are only comparing the rHla and rHla_rHlb treatment with the control PBS.  They shoud be comparing the two treatments (with and without Hlb) with each other not with PBS control.  That is the main point of the figure.  Of course oligomerization in presence of Hla will be more than PBS, where there is no Hla to oligomerize at all. 

Figure 2 needs a control that induces calcium independent of Hlb treatment to show the specificity of inhibition of Hla-elicited Ca flux by Hlb treatment.

Author Response

The authors are presenting data that shows depletion of sphingomyelin by S. aureus beta hemolysin (a sphingomyelinase) prevents the toxic activity of alpha hemolysin. The authors propose that sphingomyelin is involved in oligomerization of Hla on the surface fo target cells.  The basic data supporting this notion is ralitively weak.  Data shown in FIgure 1 are the basis for this conclusion which is the primary novelty of the paper.  however the data shown are of poor quality.  The heptamer band is poorly visible.  to make this central point of the paper more convincing the authors need to provide data of higher quality.  

Response: To avoid any problems with the densitometric quantification of Western blot bands it is advisable to keep the exposure time of the detection chip in the gel documentation as low as possible. This is not an indication of poor data quality! The data presented in Figure 1 are original data without contrast amplification or other manipulations. The data show a lack of rHla-heptamer formation upon Hlb pre-treatment, thus, it is obvious and natural that the heptamer band at the early time points is poorly visible.

Also in Fig. 1G/H/I the authors are only comparing the rHla and rHla_rHlb treatment with the control PBS.  They shoud be comparing the two treatments (with and without Hlb) with each other not with PBS control.  That is the main point of the figure.  Of course oligomerization in presence of Hla will be more than PBS, where there is no Hla to oligomerize at all. 

Response: We thank the reviewer for this hint and have now added the significance indicator to the graph. However, the reduction in heptamer formation by rHlb pre-incubation is not as pronounced as that in cultured cells. A possible explanation is that freshly isolated nasal tissue contains cells of different epithelial as well as of non-epithelial character of which some may respond differently to rHlb/rHla-treatments or not at all. This may attenuate the response of the tissue to rHlb pre-treatment somewhat when compared to that of immortalized airway epithelial cells.

Figure 2 needs a control that induces calcium independent of Hlb treatment to show the specificity of inhibition of Hla-elicited Ca flux by Hlb treatment.

Response: We have added the Supplementary Figure 2 showing that acute treatment of Indo-1 loaded 16HBE14o- cells with 5000 ng/ml rHlb does not change [Ca2+]i in the long run. This indicates that the effects of rHlb-pre-treatment on calcium signaling shown in Figure 2 are not due to indirect effects like rHlb-mediated emptying of intracellular calcium stores. A sentence explaining this has been added to the results section.

Round 2

Reviewer 2 Report

The authors have addressed some of the concerns raised in my initial review, but due to mis-interpretation of (or lack of clarity in) some of my comments, my major concern remain unaddressed.

Major Concern:

As the authors data shows, addition of a SMase (e.g. Hlb) converts all exposed sphingomyelin to ceramide. This induces profound changes in the cellular membrane via two biophysical mechanisms. The small headgroup of ceramide (compared to sphingomyelin) induces negative curvature of the membrane, resulting in the spontaneous endocytosis of a large portion of the plasma membrane independently of canonical endocytic signaling. This is a universal process across all cell types and could confound measurements of monomer binding and of calcium flux through internalization of these proteins. In addition, ceramide has a much greater propensity to co-cluster with proteins and thus inducing protein aggregation within the plane of the membrane. This has been shown - including in the 16HBE14o- cells used in this study - to alter ion channel function on the plasma membrane. Ergo, the binding measurements of Hla and the calcium flux experiments could be confounded by these processes.

Supplemental Figure 2 does not address this concern, as the concern is that Hlb would render the cell unable to induce a calcium flux, which would then produce an erroneous interpretation of the data in Figure 2. The authors should repeat the experiment in supplemental figure 2, but using an inducer of extracellular calcium flux after Hlb treatment. This would demonstrate that any cell surface ion channels Hra may co-opt remain functional. Moreover, as the protocol is written, it sounds like the experiment was conducted in PBS. If so, the experiment would not work as PBS lacks (and will precipitate) calcium ions, thereby precluding an extracellular flux.

Minor Concern:

The reviewer’s response to my previous minor comment #3 (regarding diffusion and catalytic rate issues) makes no sense. Agar is gel of sufficient density to prevent fluids from flowing through the polymer matrix. Ergo, it is not possible to "wash" materials into the gel. Indeed, this is the very principal by which a number of common assays function (Kirby-Bauer, Epsilometry, etc), and the diffusive process is well established (Fick's law with D′, equal to D(1 - αφ)/(1 - φ)). Regardless, as the data is described, no changes are necessary.

Reviewer 3 Report

The authors have minimally but adequately responded to the critiques.

Author Response

Spell check has been performed.

Round 3

Reviewer 2 Report

The authors may want to more carefully consider their replies to future reviews, to ensure that they don't literally end up lecturing (often incorrectly) someone who studies lipid membranes for a living on how membranes work. Regardless, my major issue has been addressed.